# Sequencing and assembly of the Egyptian buffalo genome

**Dina A. El-Khishin**[1]*, **Amr Ageez**[1,2], **Mohamed E. Saad**[1,3], **Amr Ibrahim**[1], **Moustafa Shokrof**[4,5], **Laila R. Hassan**[6], **Mohamed I. Abouelhoda**[7]

**1** Agricultural Genetic Engineering Research Institute (AGERI), Agricultural Research Center (ARC), Giza, Egypt, **2** Faculty of Biotechnology, MSA University, October City, Egypt, **3** Department of Biology, Taibah University, Almadinah Almonawarah, KSA, **4** Centre for Informatics Sciences, Nile University Giza, October city, Egypt, **5** Department of Computer Science, University of California at Davis, Davis, CA, United States of America, **6** Animal Production Research Institute, Agricultural Research Center (ARC), Ministry of Agriculture and Land Reclamation, Giza, Egypt, **7** Systems and Biomedical Engineering Department, Faculty of Engineering, Cairo University, Giza, Egypt

* dina_elkhishin@yahoo.com

**Data Availability Statement:** The Egyptian water Buffalo whole-genome shotgun project has been deposited at the GenBank under accession PRJNA267486. Raw DNA and RNA sequencing reads have been submitted to the NCBI Sequence

## Abstract

Water buffalo (*Bubalus bubalis*) is an important source of meat and milk in countries with relatively warm weather. Compared to the cattle genome, a little has been done to reveal its genome structure and genomic traits. This is due to the complications stemming from the large genome size, the complexity of the genome, and the high repetitive content. In this paper, we introduce a high-quality draft assembly of the Egyptian water buffalo genome. The Egyptian breed is used as a dual purpose animal (milk/meat). It is distinguished by its adaptability to the local environment, quality of feed changes, as well as its high resistance to diseases. The genome assembly of the Egyptian water buffalo has been achieved using a reference-based assembly workflow. Our workflow significantly reduced the computational complexity of the assembly process, and improved the assembly quality by integrating different public resources. We also compared our assembly to the currently available draft assemblies of water buffalo breeds. A total of 21,128 genes were identified in the produced assembly. A list of milk virgin-related genes; milk pregnancy-related genes; milk lactation-related genes; milk involution-related genes; and milk mastitis-related genes were identified in the assembly. Our results will significantly contribute to a better understanding of the genetics of the Egyptian water buffalo which will eventually support the ongoing breeding efforts and facilitate the future discovery of genes responsible for complex processes of dairy, meat production and disease resistance among other significant traits.

## Introduction

Water buffalo (*Bubalus bubalis*) is an important source of red meat and milk in countries with relatively warm weather, like India, Egypt, Brazil, Italy, and Turkey. The water buffalo population is increasing in these countries, because of its superiority to cattle in terms of higher meat and milk production, more adaptability to harsh environmental conditions, and resistance to

Read Archive database under the same project
number and sample ID (SRS750279). Links to
these data are also available on our project website
https://buffalo.tavaxy.org/.

**Funding:** This work was supported by the Science
and Technology Development Fund (STDF)
Ministry of Scientific Research, Cairo, Egypt;
Project number 2665 to Prof. Dina El Khishin,
www.stdf.eg No the The funding agency had no
role in study design, data collection and analysis,
decision to publish, or preparation of the
manuscript.

**Competing interests:** The authors have declared
that no competing interests exist.

diseases. The water buffalo (*Bubalus bubalis*) belongs to the Bovidae family, Bovinae subfamily, and Bovini tribe. Among others, the tribe Bovini contains the geneses *bos*, *Syncerus*, and *Bison*. The species *Syncerus* includes the African buffalo (*Syncerus caffer*), and the species *Bison* includes the American bison (*Bison bison*). The species *Bubalus bubalis* is subdivided into two major subspecies: The river buffalo that has 50 chromosomes and swamp buffalo that has 48 chromosomes [1–3].

The water (river) buffalo has been first domesticated in the Indian subcontinent. Then it spread to northern Africa (mainly in Egypt) and to southern Europe. With human migration water buffalo has been domesticated in Italy, Balkan and Turkey, giving them unique characteristics that differs from Egyptian water buffalos [4–8]. This suggests that the Egyptian breed is an intermediate breed between the Indian (Eastern) and European (Italian) ones [2]. The Egyptian water buffaloes are dual purpose animals used for (milk and meat). Local Egyptian breeds were shown to be more efficient when crossed with either the Italian or Pakistani buffaloes under Egyptian local conditions, proving that Egyptian breeds are more adaptive to our managerial practices, climatic and feed changes. Also, they are more resistance to diseases [9,10].

Compared to the cattle genome that is sequenced and well-characterized [11–13], little has been done to reveal the genome structure of the water buffalo. This is in spite of its importance as a major member of the Bovidae family and in spite of its interesting physiological traits related to immunology, lactation, and ruminant physiology [14]. The lack of genomic sequences limited further advanced studies addressing functional and evolutionary aspects as well as breeding optimization.

Like any other mammalian genome, the sequencing of the buffalo is complicated by the large genome size, the complexity of the genome, and its high repetitive content. To overcome these obstacles, Next Generation Sequencing (NGS) technology was used to yield a large number of DNA fragments covering the genome with a reasonable depth suitable for the assembly. A number of projects have been recently launched worldwide to sequence the genome of different buffalo breeds. In addition to the one described here, there are large-scale sequencing efforts to sequence the breeds of Italian, Indian, Bangladeshy as well as the African buffalo [15–19].

This manuscript describes the first draft assembly of Egyptian water buffalo genome. The sequencing endeavor was based on multiple runs and libraries of different insert sizes using Sequencing by Oligo Ligation and Detection (SOLiD) technology (Thermo Fisher). The assembly was achieved using a reference-based assembly workflow, highly tuned for SOLiD data. This workflow dramatically reduced the computational complexity of the assembly process and improved the assembly quality by integrating different public resources. The workflow produced an assembly with a scaffold N50 of 3.579 Mbp. Additional analyses, including the genetic content and the identification of milk genes, were also conducted. Our results, together with the other water buffalo assemblies, will significantly help in understanding the genetics of the water buffalo and opens the door to a better understanding of disease susceptibility and to further applications related to the increase of milk and meat production.

## Materials and methods

### Chosen animal

An inbred, female Egyptian water buffalo (Hathour) from the Beheiry region/type was selected for sequencing. To confirm the homozygosity level across the genome and to simplify the genome assembly, the animal was chosen after confirming half-brother and half-sister mating for three generations according to the records kept at Animal Production Research Institute

(Kafr El-Shiek, Egypt). Blood samples were collected by a competent veterinary surgeon at her home farm. The experiments in this study were performed in compliance with the official decree of the Ministry of Agriculture in Egypt relevant to animal welfare No. 27/1967 regarding the humane treatment of animals [20]. Ethics approval was obtained from the Ethics Committees of The Agricultural Genetic Engineering Research Institute and Animal Production Research Institute.

## Library preparation

DNA was extracted from whole blood using the QIAamp DNA Blood Mini Kit (Qiagen, Hilden, Germany). The buffalo genomic DNA was used to prepare short fragment libraries and mate-paired libraries. Fragment libraries were prepared by shearing two micrograms of DNA to generate fragments of approximately 100–250 bp using the Covaris S2 system (Covaris Inc., Woburn, MA, USA). The amount of the sheared DNA was confirmed by Qubit™ dsDNA HS Assay Kit (Thermo Fisher Scientific, Waltham, MA). Fragment library was prepared with the SOLiD™ fragment library construction kit (Thermo Fisher Scientific), and the SOLiD™ fragment library standard adaptors kit (Thermo Fisher Scientific) following the manufacturer's instructions. Mate-paired libraries were prepared by shearing 5–10 μg of DNA to 1 Kb and to 3 Kb fragments by the Covaris S2 system. Two mate-paired libraries, 1 Kb and 3 Kb, were generated using the 5500 SOLiD™ Mate-Paired library kit (Thermo Fisher Scientific).

## Library amplification and sequencing

Each library was amplified using emulsion PCR. 35 femto moles or 20.4 billion molecules (~5–6 ng) of a fragment library were emulsified with mineral oil in a SOLiD EZ Bead Emulsifier according to the E120 scale protocol (Thermo Fisher Scientific). The emulsion PCR was carried out in a SOLiD EZ Bead Amplifier (Thermo Fisher Scientific) using the E120 setting. To enrich for the beads that carried amplified template DNA, the beads were purified on a SOLiD EZ Bead Enricher using the recommended chemicals and software (Thermo Fisher Scientific). The beads were then purified, quantified, deposited, and attached to a sequencing flowchip. The flowchip was mounted on the 5500xl sequencer and the library was sequenced. The short fragment library was sequenced using the settings and recommended chemicals for sequencing 75 nucleotides in the forward direction and 35 nucleotides in the reverse direction, with Exact Call Chemistry (ECC) module (Thermo Fisher Scientific). Mate-Paired libraries were sequenced using, 2 x 60-bp reads were on a SOLiD 5500xl instrument, with Exact Call Chemistry (ECC) module (Thermo Fisher Scientific).

## Genome assembly

Fig 1 shows the basic steps of the analysis workflow (pipeline). Both Single-end reads and mate-paired reads were mapped to the cattle reference genome (Bos_taurus_UMD_3.1.1). The bowtie software [21] was used to map the highly similar reads to the genome. The SHRiMP software [22], which takes color space into account, was used to map the less similar reads to the reference genome. The contigs were constructed by stitching the reads mapped to the same cattle locus and overlapping with at least 50 bps. The reads that could not be mapped with Bowtie or SHRiMP were collected and assembled separately using the *de novo* program velvet [23].

The mate-paired reads were then used to confirm the assembled contigs, and to assemble the contigs into larger scaffolds. A contig is confirmed if the two parts of the paired-end reads are mapped to it. Paired-end reads mapped to two different contigs are used in scaffolding the

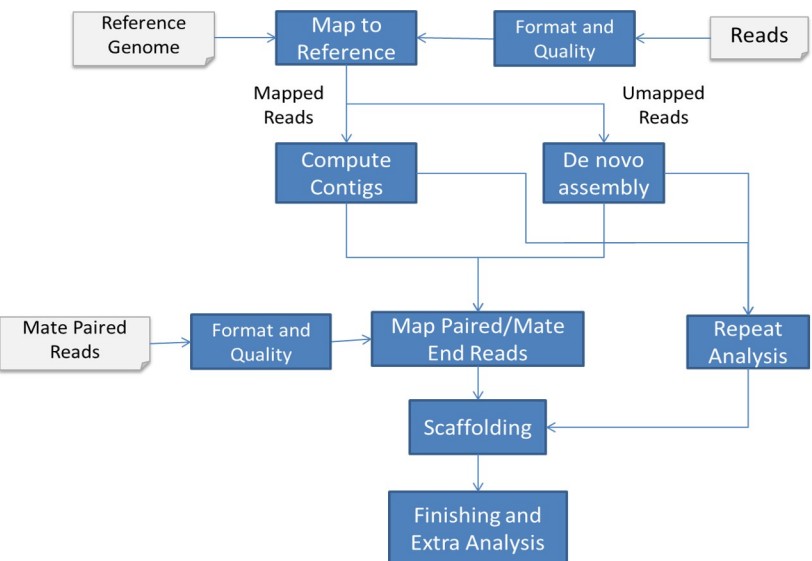

**Fig 1. Water buffalo genome assembly workflow.** Both mate-paired and short fragment reads, after trimming at Q15, were mapped to the cattle genome. The mapped reads were used to construct contigs. The un-mapped reads were collected and assembled separately using the de novo program velvet. The mate-paired reads were used to confirm assembled contigs, and to assemble the contigs into larger scaffolds. The greedy path-merging algorithm, Opera, Sopra, and Grass algorithms were used in the scaffolding process.

two contigs. The greedy path-merging algorithm, like the one implemented in Opera, Sopra, and Grass [24–27] was used in the scaffolding process.

The S1 File includes more details about the implementation of the assembly workflow, which is based on the Tavaxy Workflow Management System [28] and the distributable version available as Docker image. Web links to the workflow implementation is also available at the project website https://egyptianbuffalo.org/ and https://buffalo.tavaxy.org.

## Genome annotation

**Repeat analysis.** RepeatMasker [29], with default parameters, was used to locate the repeats in the assembly. The resulting output was analyzed using in-house scripts to compute coverage of repeats, classify them into families, and produce final statistics and charts.

**Gene analysis.** A pipeline similar to the NCBI Eukaryotic Genome Annotation Pipeline [30], was used to annotate the assembly. The coding genes in the Egyptian buffalo genome were identified over three phases. During the first phase, the mammalian mRNA sequences available in the NCBI were aligned to the draft genome. The mammalian mRNA sequences included RefSeq mRNA data from cattle, human, mouse, rat, and pig. The alignment of mRNA to the Egyptian water buffalo genome was achieved using BLAT [31], with a cross-species setting of 80% identity and a minimum hit length of 50b. The second phase was performed using homology-based prediction by transferring the cattle's annotation to our assembly. The program GASS [32] was then used to map the gene sequences to the draft genome. In the final stage, extra genes were identified using the de novo prediction Augustus program [33].

**Comparison to other assemblies.** The assembly of each draft genome was collected from the NCBI Assembly database. Each assembly was then processed using RepeatMasker [29] to mask out the repeated regions. The alignment between the different assemblies was achieved using Minimap2 [34], CoCoNUT [35] and Mauve [36], where the minimum length of a reported alignment is 100bp. Colinear alignments were coalesced into larger syntenic blocks.

To facilitate the plotting of the synteny blocks, we used the chromosome-level assembly of the cattle and UOA buffalo genomes as references. The plots were generated using the Rideogram package in R.

## Results and discussion

In this paper, the Egyptian water buffalo genome was sequenced using the SOLiD technology. To the best of our knowledge, this is the first mammalian genome to be sequenced using this technology. Two-base encoding offered by the SOLiD system is critical for robust detection of sequencing errors to produce highly accurate base calls necessary for genome sequencing [37]. Reference-based assembly strategy was used to produce the assembly, this strategy can overcome the short read length produced by this sequencing platform, speed up the computation, and produce a high-quality assembly. Specifically, the mapping step including the color-space based SHRiMP program [22] could map most of the data to the reference genome and facilitated the assembly of the contigs. The use of *de novo* assembler was then limited to a much-reduced set of unmapped reads. The reported assembly has been verified using different information sources, including mate-paired reads and gene sequences from public databases.

The incorporation of sequences from both mate-paired and single-read libraries significantly improved the production of long scaffolds. In addition to their importance in genome annotation, the use of publicly available gene sequences from related organisms also helped to verify the genome assembly and improve its quality by correcting some scaffolds.

### Sequencing

For sequencing, a female Egyptian water buffalo was selected from the Beheiry region/type. DNA was extracted from peripheral venous blood, and two short fragment libraries, one 1 Kb mate-paired, and one 3 Kb mate- paired library were constructed. The genome was sequenced using the SOLiD DNA sequencing platform (SOLiD 5500XL sequencer). In total, 3.4 billion reads with 229.24 Gb were generated, achieving about 80X-fold coverage of the water buffalo genome. After quality trimming at Q15, 2028 million reads (~60%) of total length 100.85 Gb passed the filtering, achieving about 36-fold coverage of the water buffalo genome. Table 1 summarizes the resulting sequencing data and related quality metrics. The reads of different runs were submitted to NCBI under project number PRJNA267486, and a copy of it is accessible from our website http://egyptianbuffalo.org.

### Short-read *de novo* sequencing and assembly

The river buffalo genome has 5 bi-armed and 19 acrocentric chromosome pairs in addition to the XY sex chromosomes [14,38]. The water buffalo genome was assembled using a reference-based assembly workflow using the "Tavaxy" workflow management system [28]. Tavaxy is an

**Table 1. Four sequencing runs statistics.** No trimming was made on Q0 and all bases are included. Q15 means trimming the bases and the ones after them if their quality drops below 15. Reads shorter than 30 bps are discarded. Run 3_1K and Run 4_3K include mate-paired reads with an insert size of 1 Kbp and 3Kbp, respectively.

| Runs | #Reads at Q0 (in million) | #Reads at Q15 (in millions) | #Read bases (yield) at Q0 (in GBp) | #Read bases (yield) at Q15 (in Gbp) | Average/Median at Q0 (bp) | Average at Q15 (bp) | Median at Q15 (bp) |
|------|---------------------------|------------------------------|-------------------------------------|--------------------------------------|----------------------------|----------------------|---------------------|
| Run1 | 880 | 565 | 66 | 28.815 | 75 | 51 | 52 |
| Run2 | 778 | 546 | 58.3 | 28.938 | 75 | 53 | 55 |
| Run3_1K | 944 | 517 | 56.64 | 24.299 | 60 | 47 | 50 |
| Run4_3K | 805 | 400 | 48.3 | 18.8 | 60 | 47 | 47 |
| **Total** | **3407** | **2028** | **229.24** | **100.852** | | | |

extended version of Galaxy enabling the execution of the workflow using cloud computing. The use of reference-based assembly workflow was preferred over the *de novo* assembly workflow due to the short read length produced by the SOLiD platform; which will not be easily assembled into longer contigs using *de novo* assembler. Furthermore, the abundance of repeats in the genome leads to an ambiguity that cannot be solved in the assembled regions using short reads/contigs.

The cattle genome (Bos_taurus_UMD_3.1.1) was selected as the reference genome. At the chromosomal level, the reads of the Egyptian buffalo covered all chromosomes with an average of 98.1%. Also, the assembled contigs from the mapped reads covered 97.5% of the cattle genome (S2 File) these primary results justified the use of the cattle genome as a reference for the assembly.

## Assembly results

Raw reads were successfully assembled into 447,289 contigs with a total length of 3.005 Gb (Table 2). The maximum contig length was 148 Kb, and the N50 of contigs was 14.568 Kb. The number of contig lengths larger than 1 Kb was 298,457 with a total length of about 2.475 Gb. Mate-paired read libraries of different lengths (1 Kb and 3 Kb) have been used to confirm the contigs. A contig is considered verified if both ends of at least three mate-paired reads are mapped to it and there are no mate-paired reads connecting this contig to another one in a non-suffix prefix manner.

The scaffolds have been confirmed using mate-paired read libraries of different lengths (1 Kb and 3 Kb), and also by aligning the publically available cattle mRNA genes to the draft assembly. Out of the tested 11653 cattle genes, 11138 genes were mapped correctly with complete conservation of the gene structure within one scaffold.

## Comparison to other buffalo assemblies

A comparison of the Egyptian buffalo to all available buffalo assemblies deposited in the NCBI database until June 2020 was performed. These assemblies included the Mediterranean UOA_WB assembly [17], the Bangladeshy assembly [18], the Mediterranean UMD_Caspur [15,39], the Indian [19], and the African buffalo assembly [16] (Table 3). All new assemblies

**Table 2. Assembly metrics of the Egyptian water buffalo genome.**

| Contig Statistics | Value |
|---|---|
| Total sequence length | 3,005,952,034 |
| Total ungapped length | 2,677,178,324 |
| Gaps between scaffolds | 0 |
| Number of scaffolds | 99,165 |
| Scaffold N50 | 3,579,700 |
| Scaffold L50 | 241 |
| Number of contigs | 447,289 |
| Contig N50 | 14,568 |
| Contig L50 | 54,293 |
| Total number of chromosomes and plasmids | 0 |
| Number of component sequences (WGS or clone) | 447,289 |

The scaffolding process yielded 99,165 scaffolds with an estimated total length of 2.8 Gb. The N50 of the assembled genome is ≈3.58 Mb, with maximum and average scaffold lengths of 20.3 Mb and 608.4 Kb, respectively.

**Table 3. A. Information about the assemblies of Water and African buffalos deposited in NCBI. B.** Comparison between the Egyptian buffalo assembly and other buffalo assemblies using different assembly metrics.

**A**

| Water Buffalo (*Bubalus bubalis*) | | | | | |
|---|---|---|---|---|---|
| Assembly | Date | Breed | GenBank ID | Project ID | Seq. Technology |
| UOA_WB_1 (RefSeq) | Jan 2019 | Mediterranean | GCA_003121395.1 | PRJNA437177 | PacBio |
| Bubbub1.0 | Apr 2019 | Bangladesh | GCA_004794615.1 | PRJNA349106 | Illumina HiSeq 2000 |
| ASM299383v1 | Mar 2018 | Egypt | GCA_002993835.1 | PRJNA267486 | SOLiD |
| Bubalus_bubalis_Jaffrabadi_v3.0 | Feb 2018 | Jaffrabadi | GCA_000180995.3 | PRJNA40113 | 454; Illumina NextSeq 500 |
| UMD_CASPUR_WB_2.0 | Sep 2013 | Mediterranean | GCA_000471725.1 | PRJNA207334 | Illumina GAIIx; Illumina HiSeq; 454 |
| African Buffalo (Syncerus caffer) | | | | | |
| Synceruscaffer1 (EDINBURGH/Aber) | Apr 2020 | African | GCA_902825105.1/GCA_902500845.1 | PRJEB36586 | PacBio |
| ABF | Jun 2019 | African | GCA_006408785.1 | PRJNA438286 | Illumina HiSeq |

**B**

| Water Buffalo (*Bubalus bubalis*) | | | | | | | |
|---|---|---|---|---|---|---|---|
| | Level | Length | Scaffold Count | Contig Count | Ungapped Length | Scaffold N50 | Contig N50 |
| **UOA_WB (RefSeq)** | Scaffold + Chromosome | 2,655,780,776 | 509 (26 Chrom) | 919 | 2,655,407,276 | 117,219,835 | 22,441,509 |
| **Bangali Bubbub1.0** | Scaffold | 2,770,260,352 | 33,821 | 235,948 | 2,630,018,892 | 6,957,949 | 25,038 |
| **Egypt:** | Scaffold | 3,005,952,034 | 99,165 | 447,289 | 2,677,178,324 | 3,579,700 | 14,568 |
| **UMD_CASPUR:** | Scaffold | 2,836,150,610 | 366,982 | 630,367 | 2,761,762,569 | 1,412,388 | 21,938 |
| **Jaffrabadi:** | Scaffold | 3,759,980,894 | 117,845 | 432,941 | 2,909,441,936 | 102,345 | 13,977 |
| African Buffalo (*Syncerus caffer*) | | | | | | | |
| **Edinburgh/Aber (RefSeq)** | Scaffold | 2,652,966,730 | 3,265 | 6,978 | 2,649,984,299 | 69,160,875 | 2,018,310 |
| **ABF** | Scaffold | 2,929,427,825 | 148,371 | 658,175 | 2,869,481,663 | 2,316,376 | 42,601 |

appeared in 2019 and 2020, and the UMD water buffalo and the African buffalo assemblies were updated within the last year using the recent long read sequencing methods.

Comparisons of the Egyptian buffalo to these water and African buffalo assemblies as well as to the cattle genome were performed (Table 3B). At the contig level, the Egyptian assembly has a moderate N50 compared to other *Bubalus bubalis* draft assemblies, which is expected due to the short-read output of the SOLiD technology. Comparing the scaffolds, the assembly of the Egyptian buffalo had N50 of 3.57 MB which is the second largest N50 among the *Bubalus bubalis* draft assemblies based on short-read sequencing. The usage of mate-paired read libraries, aligning the publically available cattle mRNA genes to draft assembly, and the usage of UMD assembly are the main reasons for refining our assembly.

Irrespective of the differences in the number of scaffolds, the Egyptian draft assembly is in good agreement with the Mediterranean buffalo (UOA and UMD assemblies). At the read level, the reads of the draft Egyptian assembly covered 99.59% of the UMD/UOA assembly. At the scaffolding/contig level, 99.3% of the Egyptian buffalo assembly is mapped to the UMD/UOA assembly.

The synteny blocks among the draft assemblies were computed, taking the cattle and UOA assemblies as references. The cattle and UOA genomes are currently the reference genome assemblies: They were assembled up to the level of chromosome and they were intensively verified using different support methods like Hi-C, RH and optical mapping. Fig 2 shows different plots of the synteny blocks with the UOA and Cattle genomes as references. The plots are based on plotting syntenic blocks of at least 1Mbp, except for the Indian buffalo where we had to reduce the threshold to 500Kbp. As expected, the new UOA Mediterranean assembly [17] has better placement of scaffolds and chromosomal structure. It is in high synteny with the cattle genome. It can also be observed that the assemblies based on short read sequencing, notably the Bangaladeshi and ours, could achieve good level of synteny as did the new UOA assembly.

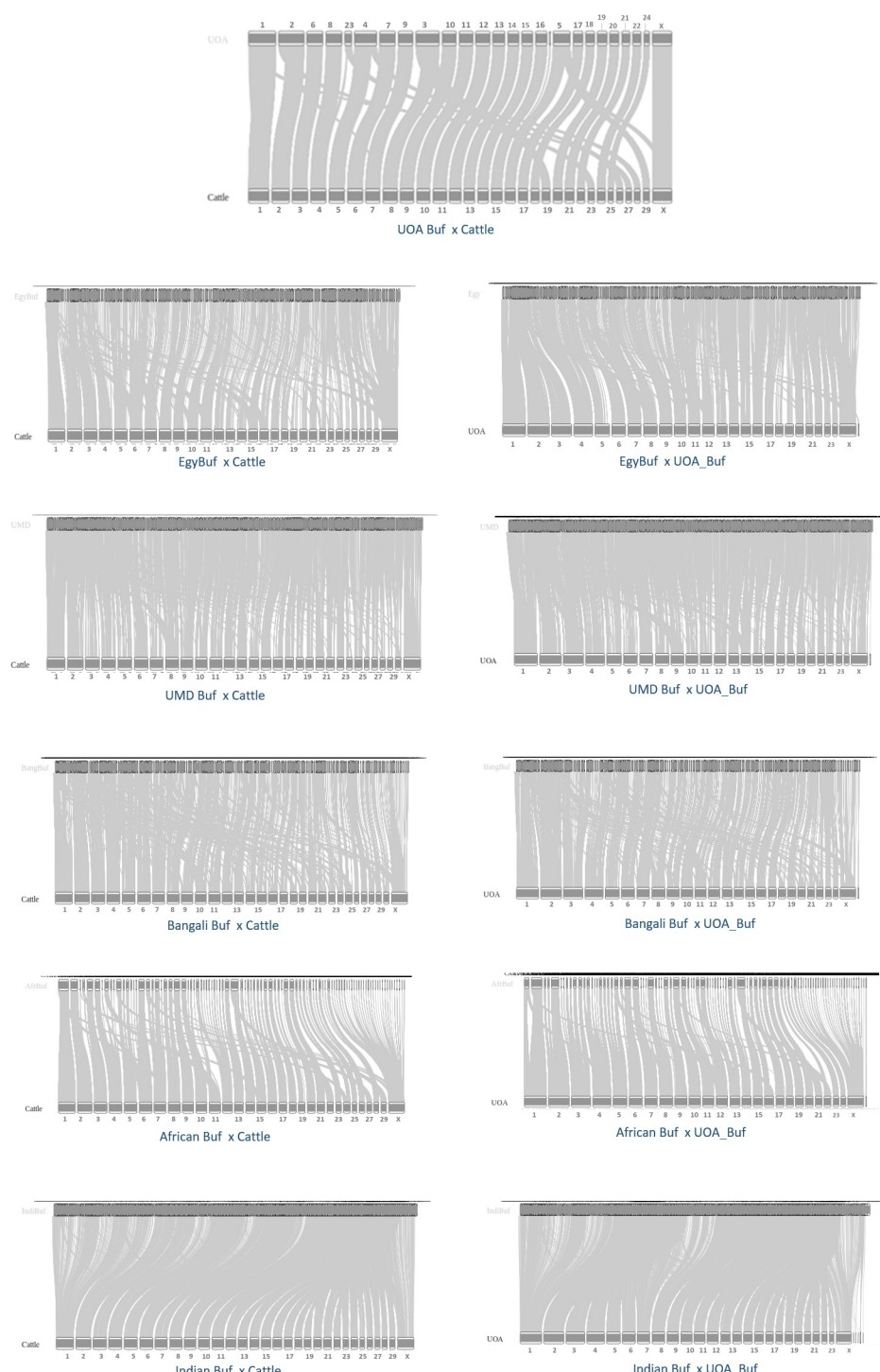

**Fig 2. Synteny blocks between different buffalo assemblies and the UOA and cattle assemblies.** The first plot on the top includes the synteny diagram between the well revised genomes of the UOA buffalo genome and Cattle. The plots are based on plotting syntenic blocks of at least 1Mbp, except for the Indian buffalo where we had to reduce the threshold to 500Kbp.

**Table 4. Repeat content (in bp) in the cattle genome and Egyptian water buffalo genome.** ncRNA includes rRNA, scRNA, snRNA, srpRNA, and trRNA.

| Repeat Family | Cattle genome | Egy Buffalo genome |
|---|---|---|
| LINE | 591242926 | 564818087 |
| LTR | 125345371 | 115039365 |
| SINE | 455612491 | 443033730 |
| Simple repeat | 20557705 | 16644633 |
| Satellite | 7090776 | 2966452 |
| ncRNA | 21196503 | 5714700 |
| Others | 55795102 | 53967410 |
| Total | **1,276,840,874** | **1,202,184,377** |

Future work to enhance these assemblies using other runs of long read assemblies would definitely lead to significant improvement in scaffolding.

## Genome annotation

### Repeat analysis

Repeats in the Egyptian water buffalo genome assembly were identified using the program RepeatMasker [29]. Analysis of the repeats revealed that a 1.2 Gb, 39.3% based upon genome size 3.05 Gb, of the Egyptian water buffalo genome is composed of repeats. The repeat content in Egyptian water buffalo assembly is less than that of the cattle genome (1.27 Gb). The LINE family comprises around 49% of the repeats (46% in the cattle), followed by the SINE family representing 36.8% (35.6% in the cattle). LTR makes up 10% in both water buffalo and cattle genomes (Table 4).

### Gene content

Annotation of genes, transcripts, was performed using the NCBI Eukaryotic Genome Annotation Pipeline [30]. The annotation of the Egyptian water buffalo genome yielded 21128 genes (S3 File). Of these, 11,702 ones came directly from the mammalian mRNAs. A total of 9426 genes that have no mRNA hit were identified using Augustus annotations. A total of 3753 pseudogene loci from 603 genes were identified (S4 File). In addition, there were 5393 non-coding RNAs (1202 rRNA, 2242 tRNA, and 1949 other small RNAs), identified by RepeatMasker.

**Milk genes.** Buffalo milk accounts for 13% of the total world milk production (http://faostat.fao.org/). Although the buffalo provides much lower average milk yield than that of Holstein cows [40], its milk is one of the best raw materials for making dairy products [41]. Buffalo milk has higher fat content, higher unsaturated fatty acids, higher milk protein and lower levels of cholesterol compared to that's of the cow's milk [42].

Although milk composition is variable across species, the milk and mammary genes are more conserved in most mammals than are other genes in the genome. Therefore, milk genes were identified in our assembly using the annotation of milk lactation gene sets [13]. Collectively, 3889 e milk virgin-related genes; 1,383 milk pregnancy-related genes; 3,111 milk lactation-related genes; 867 milk involution-related genes; and 840 milk mastitis-related genes were identified inside the Egyptian draft assembly (Fig 3 and S5 File).

### Pathway and gene ontology analysis of milk genes

The molecular functions of milk virgin-related genes; milk pregnancy-related genes; milk lactation-related genes; milk involution-related genes; and milk mastitis-related genes were

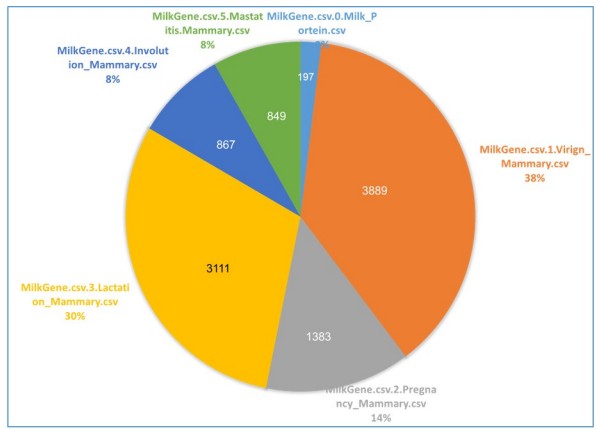

**Distribution of Genes in Milk and Mammary Sets**

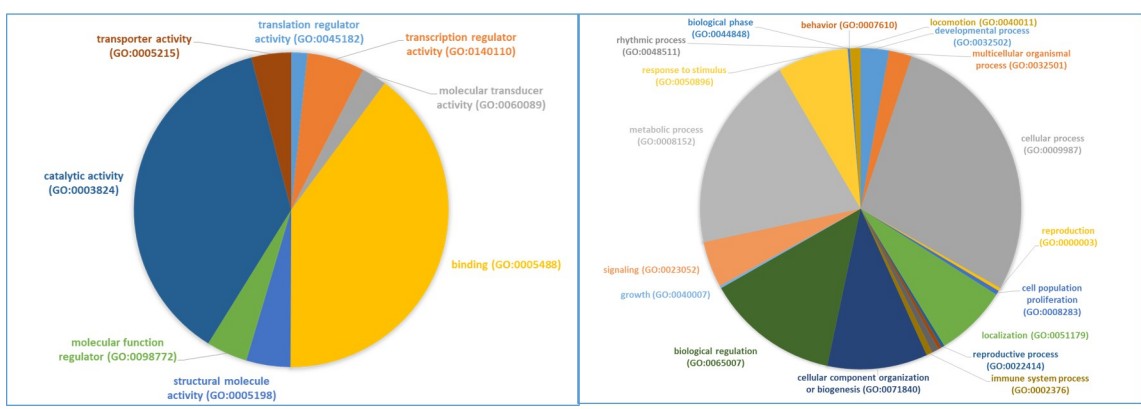

**GO: Molecular Function**

**GO: Biological Process**

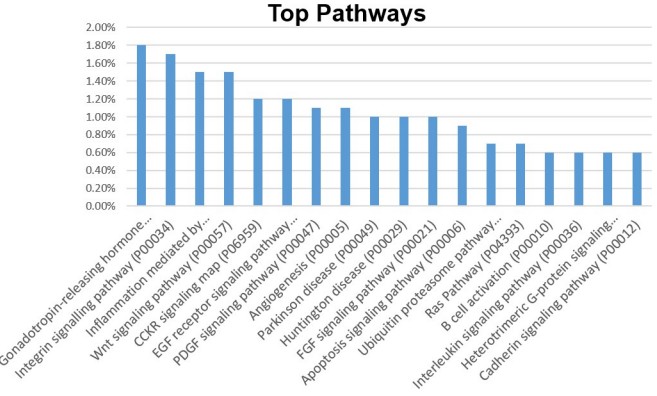

**Fig 3. The top part includes a chart for the number of genes in different milk and mammary gene sets.** The middle part includes gene ontology analysis: molecular function (left side) and biological processes (right side). The bottom part includes top pathways related to all genes in the milk and mammary sets.

mapped using PANTHER pathway analysis kit [43]. Each milk lactation gene-set was classified according to their possible involvement in molecular functions, biological process, cellular

component, and different pathways (Fig 3 and S6 File). The identified genes encompass a wide spectrum of molecular functions, biological processes, pathways, cellular components and Panther protein classes. The genes were successfully annotated and classified into the following categories: binding, proteins with different catalytic activity, regulators of molecular function, molecular transducer activity related proteins, structural molecule activity, transcription regulator activity, translation regulator activity, and transporter activity.

## Conclusions

The availability of the Egyptian buffalo genome is an addition to the global efforts for unravelling the genomic basis of the buffalo biology. The paper at hand introduces a high quality draft assembly based on short-read sequencing using the SOLiD platform. To the best of our knowledge, this is the first large mammalian genome to be sequenced using this technology. Combining libraries of single and paired-end reads were crucial to refine the assembly and produce long scaffolds. The raw sequences and the assembled contigs and scaffolds have been made public and deposited at the NCBI repository. Genome annotation was conducted and gene lists have been produced. We focused our analysis on milk genes and analysed the respective pathways and gene ontology.

The usage of modern long-read sequencing has shown great advantage in producing better scaffolds, as demonstrated in the recent Italian and African buffalo assemblies and the African buffalo. We plan to use similar technology in the future to improve the scaffolding and provide well-phased reference assembly for the Egyptian breed.

We expect that the next logical step in buffalo genome research is to run large scale genome variation projects on a large number of animals and to conduct association studies to predict genes and variations contributing to the unique characteristics of the buffalo breeds. This step will make use of the available well-phased reference assemblies to efficiently compute these variations.

The Egyptian water buffalo genome offers a unique insight for understanding the variations in sequence between buffalo around the world and cattle imporarant genes. This variation could help in the future discovery of genes behind complex dairy and meat production traits and to understand the species-specificity of milk composition. Moreover, the availability of various buffalo assemblies with gene annotation will offer a good opportunity for understanding the disease resistance/susceptibility in buffaloes. This work will eventually help in identifying and validating specific genes related to the quantitative and qualitative aspects of milk production. Such tools will enable the farmer to predict the characteristics of the desired animal and be able to invest in appropriate animals with higher milk or meat production without wasting effort, time and money.

## Supporting information

**S1 File. Description of implementing the assembly workflow on the Tavaxy workflow management system.**
(DOCX)

**S2 File. Buffalo reads and contigs mapping against the cattle genome (detailed view per cattle chromosomes).**
(DOCX)

**S3 File. List of genes identified in the Egyptian draft assembly.**
(XLSX)

**S4 File. List of pseudogenes identified in the Egyptian draft assembly.**
(XLSX)

**S5 File. List of milk lactation gene sets identified in the Egyptian draft assembly.**
(XLSX)

**S6 File. Classification of milk lactation gene sets according to their possible involvement in molecular functions, biological process, cellular component, and different pathways.**
(XLSX)

## Acknowledgments

We would like to thank Dr. Moustafa Ghanem, Mr. Mohamed El-Kalioby, Mr. Ahmed Ali, and Mr. Hatem Elshazly for supporting us in this project.

## Author Contributions

**Conceptualization:** Dina A. El-Khishin, Mohamed I. Abouelhoda.

**Data curation:** Moustafa Shokrof, Mohamed I. Abouelhoda.

**Formal analysis:** Moustafa Shokrof, Mohamed I. Abouelhoda.

**Funding acquisition:** Dina A. El-Khishin.

**Investigation:** Dina A. El-Khishin, Amr Ageez, Mohamed E. Saad, Amr Ibrahim.

**Methodology:** Amr Ageez, Mohamed E. Saad, Amr Ibrahim, Laila R. Hassan, Mohamed I. Abouelhoda.

**Project administration:** Dina A. El-Khishin.

**Resources:** Laila R. Hassan.

**Software:** Mohamed I. Abouelhoda.

**Supervision:** Dina A. El-Khishin.

**Writing – original draft:** Dina A. El-Khishin, Amr Ageez, Mohamed E. Saad, Mohamed I. Abouelhoda.

**Writing – review & editing:** Dina A. El-Khishin, Amr Ageez, Mohamed E. Saad, Mohamed I. Abouelhoda.

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
