## [Decision Letter · Decision Letter 0]

29 Apr 2020

PONE-D-20-01913

Sequencing and Assembly of the Egyptian Buffalo Genome

PLOS ONE

Dear Prof. El Khishin,

Thank you for submitting your manuscript to PLOS ONE. After careful consideration, we feel that it has merit but does not fully meet PLOS ONE’s publication criteria as it currently stands. Therefore, we invite you to submit a revised version of the manuscript that addresses the points raised during the review process.

The study is good work and well presented, I feel that the manuscript is dealing with a good topic but lacks in the quality of preparation. I agree with reviewers, and please review the referee comments and make your peer revision.

We would appreciate receiving your revised manuscript by Jun 12 2020 11:59PM. To enhance the reproducibility of your results, we recommend that if applicable you deposit your laboratory protocols in protocols.io, where a protocol can be assigned its own identifier (DOI) such that it can be cited independently in the future. For instructions see: http://journals.plos.org/plosone/s/submission-guidelines#loc-laboratory-protocols

We look forward to receiving your revised manuscript.

Kind regards,

Arda Yildirim, Ph.D.

Academic Editor

PLOS ONE

Journal Requirements:

3. Thank you for including your ethics statement: 

"Animal was chosen and cared for by the guidelines of Animal Production Research

Institute, Agricultural Research Center (ARC), Ministry of Agriculture and Land

Reclamation, Egypt."

Please amend your current ethics statement to confirm that your named ethics committee specifically approved this study.

For additional information about PLOS ONE submissions requirements for ethics oversight of animal work, please refer to http://journals.plos.org/plosone/s/submission-guidelines#loc-animal-research  

Additional Editor Comments (if provided):

The study is very well presented, I feel that the manuscript is dealing with a good topic but lacks in the quality of preparation. The main problem found in the manuscript is related to the some aspects of the methodology and typo errors or ambiguous phrases or sentences. It is necessary to improve the manuscript by examining the questions that need to be clarified in a way. Moreover it should be expressed as water buffalo rather than the Buffalo on whole text. Anatolian water buffaloes also the geographic region in Turkey after Italy has also reared widely and I would recommend updating the introduction and citing the literature listed below to improve and contextualize the paper. Some of the suggested papers are new but they already got a good number of citations:

Şahin, A., Yıldırım, A., Ulutaş, Z., 2014. Some physicochemical characteristics of raw milk of Anatolian Buffaloes. Italian Journal of Food Sciences, 26(4):398-404.

Şahin, A., Yıldırım, A., Ulutaş, Z., 2016. Changes in some physico-chemical content of Anatolian Buffalo milk according to the some environmental factors. Buffalo Bulletin, 35(4):573-585.

Şahin, A., Yıldırım, A., Ulutaş, Z., 2016. Effect of Various Environmental Factors and Management Practices on Somatic Cell Count in the Raw Milk of Anatolian Buffaloes. Pakistan Journal of Zoology, 48(2):325-332.

Şahin, A., Yildirim, A., Ulutaş, Z., 2017. The effects of stage of lactation, parity and calving season on somatic cell counts in Anatolian Water Buffaloes. Indian Journal of Animal Science, 51(1):35-39.

Şahin, A., Yıldırım, A., Ulutaş, Z., 2019. The Effects of storage temperature and storage time on the somatic cell count of Anatolian Buffaloes. Buffalo Bulletin. 38(2):299-309.

Please be aware of the manuscript should be presented according to guidelines for authors of Plos One. For your guidance, you can check the reviewers' comments.

Reviewers' comments:

Reviewer's Responses to Questions

**Comments to the Author**

1. Is the manuscript technically sound, and do the data support the conclusions?

Reviewer #1: Yes

Reviewer #2: Yes

Reviewer #3: Yes

2. Has the statistical analysis been performed appropriately and rigorously? 

Reviewer #1: Yes

Reviewer #2: Yes

Reviewer #3: I Don't Know

3. Have the authors made all data underlying the findings in their manuscript fully available?

Reviewer #1: Yes

Reviewer #2: Yes

Reviewer #3: Yes

4. Is the manuscript presented in an intelligible fashion and written in standard English?

Reviewer #1: No

Reviewer #2: Yes

Reviewer #3: Yes

5. Review Comments to the Author

Reviewer #1: The manuscript presented by El-Khishin and coworkers presents novel data and new insights about the Egyptian buffalo genome. In this work a high-quality draft assembly has been performed and compared compare to the currently available draft assemblies of buffalo breeds. A list of genes related to milk virgin, milk pregnancy, milk lactation, milk involution and milk mastitis have been finally identified. This is an interesting work to better understand the genetics of the Egyptian buffalo that will very probably enhance more accurate breeding programes. However, some comments need to be addressed:

1.- English style should be revised in the whole manuscript.

2.- The Ethical Committee information is lost and it should appear in the Material and Methods section.

3.- The authors indicate that the analysis performed in this study could provide the identification of a large list of genes related to milk virgin, milk pregnancy, milk lactation, milk involution and milk mastitis, albeit only information about milk lactation related genes is provided in Support Table 3. Please, provide more information about the rest of lists of genes.

4.- Additional graphs about the particular function of the identified genes in each list could help for a better understanding of the genes related to each category. I encourage the authors to make groups of genes that could be involved in different molecular and cellular functions inside each general list using, for example, cheese plots or cheese graphics.

5.- The Results and Discussion section needs to be more elaborated and include more references to support the findings.

Reviewer #2: What is the application aspect of the study?

How It is applicable to daily life of farmers?

Which sector it covers the most?

Does this study addressed scientists only?

What is the future of your research?

Reviewer #3: My first question when checking the submission of the sequence data is that it was deposited at ncbi the 27-Sep-2013.

Since then several water buffalo genomes have being published, the authors mention one (CASPUR-assembly) and use that information to make comparisons. If accepted for publication this work should include the other water buffalo genomes, eg https://doi.org/10.1002/ece3.4965

I guess that the authors used the cattle as reference genome for the assembly because there was no other water buffalo draft available at the time when this study was initiated? Can the authors explain why they used the cattle and not another water buffalo genome? Will the assembly be different? Discuss the differences.

I recommend to rewrite several parts of the manuscript to make clear what kind of buffalo species and subspecies the authors are referring to in the text (latin names). As example not all the african buffalo belongs to the same sub-species.

doi: 10.1371/journal.pone.0056235

If published I would like to have a table with the different buffalo species mentioned in the article informing about the number of chromosomes that each has. Would be very informative to have synteny figures between the Egyptian water buffalo and cattle, african buffalo, other water buffalos.

Of interest for the reader it would be good know what are the differences between the Egyptian, the "Italian" , the chinese and other published water buffalo genomes, a comparison at the genome level is desirable. In all, why the Egyptian water buffalo is worth its own genome publication.

Most publishers ask for accession numbers for the sequence data produced, I would like to have the workflow used in this work deposited somewhere so it can be studied by the reviewers e.g https://fair-workflows.github.io/project.html

There is a link in the article mentioning that the authors used a workflow using Tavaxy, but I could not find any instructions at the Tavaxy site. https://www.tavaxy.org/

I also visited the site and many links (e.g. https://egyptianbuffalo.org/data) gave:

------------

Not Found

The requested URL was not found on this server.

Apache/2.4.29 (Ubuntu) Server at egyptianbuffalo.org Port 443

------------

In my opinion the access to genomic data of many variants of a specific specie is very desirable as just the existence of one reference genome is not sufficient to understand complex biological processes. Therefore the publication of the Egyptian water buffalo Genome is of interest but only if the authors explain and correct the issuers mentioned above.

6. PLOS authors have the option to publish the peer review history of their article (what does this mean?). If published, this will include your full peer review and any attached files.

Reviewer #1: Yes: Ana Cristina Calvo

Reviewer #2: Yes: Akhtar Rasool Asif

Reviewer #3: No

---

## [Author Response · Author response to Decision Letter 0]

19 Jun 2020

Briefly, we addressed the editorial comments as follows:

- We reviewed the language throughout the paper and a native speaker helped us improve the English style. 

- We improved the introduction and included the references suggested. 

- We corrected the ethical statements as per the journal’s guidelines.

- We added extra supplementary file (S2 Supplementary File 2) for the data that was not shown in the first submission. 

- We reviewed other editorial items related to formats and organization of submitted files.

Regarding the individual’s reviewer comments, we addressed each point carefully. Briefly, we addressed the reviewers’ comments as follows:

- In response to Reviewers 1 & 2 & 3, we improved the introduction and discussion section to include more information about the origin and differences of the buffalo species and about the applicability of the project for both academia and the farm. We also included extra text to put our research in context with global buffalo/livestock research and referred to future research venues in the conclusion section, notably 

1. The use of long-read sequencing for producing high quality well-phased assembly up to the chromosome level, and

2. The initiation of large scale projects to study genomic variation on large number of animals and to conduct association studies to these variations to different traits related to milk/meat production and resistance to disease.

- In response to Reviewer 1, we conducted extra analysis for milk and mammary genes. We performed pathway and gene ontology analyses. We included the results in an additional Supplementary file (S6 Table) and added a new figure (Figure 3) in the manuscript summarizing the analysis results.

- In response to Reviewer 3, we reviewed all available assemblies in the NCBI up to June 2020. We collected all available sequences and conducted large scale analysis to compute synteny blocks. The genome comparison section was updated and a synteny plot was added (Figure 2). It is worth mentioning that many of these assemblies appeared in the time of submitting the manuscript. Two of these new assemblies used the modern long-read sequencing technology to update their assemblies (the UMD Italian Buffalo and African Buffalo), which led to significant improvements. 

- In response to Reviewer 3, we packaged our workflow for assembly in a docker image and made it publically available in DockerHub. We also reviewed all our links at the web-site and included links to the data in NCBI (runs, contigs, and assembly), and also included links to the workflow systems in DockerHub and the Tavaxy server dedicated to the Buffalo Project. In addition, we included an extra Supplementary file (S1 Supplementary File 1) including instruction how to access the Docker image and do the basic analysis steps.

'Response to Editorial Comments'

1-- Please ensure that your manuscript meets PLOS ONE's style requirements, including those for file naming. The PLOS ONE style templates can be found at https://journals.plos.org/plosone/s/file?id=wjVg/PLOSOne_formatting_sample_main_body.pdf and https://journals.plos.org/plosone/s/file?id=ba62/PLOSOne_formatting _sample_title_authors_affiliations.pdf

Response:

We reviewed these editorial items related to formats and organization of submitted files, and the manuscript has been updated accordingly.

2- We note that you have included the phrase “data not shown” in your manuscript. Unfortunately, this does not meet our data sharing requirements. PLOS does not permit references to inaccessible data. We require that authors provide all relevant data within the paper, Supporting Information files, or in an acceptable, public repository. Please add a citation to support this phrase or upload the data that corresponds with these findings to a stable repository (such as Figshare or Dryad) and provide and URLs, DOIs, or accession numbers that may be used to access these data. Or, if the data are not a core part of the research being presented in your study, we ask that you remove the phrase that refers to these data.

Response:

We added extra supplementary file (S2 Supplementary File 2.) for the data that was not shown in the first submission. 

3- Thank you for including your ethics statement: "Animal was chosen and cared for by the guidelines of Animal Production Research Institute, Agricultural Research Center (ARC), Ministry of Agriculture and Land Reclamation, Egypt." Please amend your current ethics statement to confirm that your named ethics committee specifically approved this study.

Response:

We modified the text in the manuscript to confirm the approval of this study. We re-wrote the paragraph as follows:

“The experiments in this study were performed in compliance with the official decree of the Ministry of Agriculture in Egypt relevant to animal welfare No. 27/1967 regarding the humane treatment of animals (19). Ethics approval was obtained for this study from the Ethics Committees of The Agricultural Genetic Engineering Research Institute and Animal Production Research Institute.”

4- The study is very well presented, I feel that the manuscript is dealing with a good topic but lacks in the quality of preparation. The main problem found in the manuscript is related to the some aspects of the methodology and typo errors or ambiguous phrases or sentences. It is necessary to improve the manuscript by examining the questions that need to be clarified in a way. Moreover it should be expressed as water buffalo rather than the Buffalo on whole text. Anatolian water buffaloes also the geographic region in Turkey after Italy has also reared widely and I would recommend updating the introduction and citing the literature listed below to improve and contextualize the paper. Some of the suggested papers are new but they already got a good number of citations:

- Şahin, A., Yıldırım, A., Ulutaş, Z., 2014. Some physicochemical characteristics of raw milk of Anatolian Buffaloes. Italian Journal of Food Sciences, 26(4):398-404.

- Şahin, A., Yıldırım, A., Ulutaş, Z., 2016. Changes in some physico-chemical content of Anatolian Buffalo milk according to the some environmental factors. Buffalo Bulletin, 35(4):573-585.

- Şahin, A., Yıldırım, A., Ulutaş, Z., 2016. Effect of Various Environmental Factors and Management Practices on Somatic Cell Count in the Raw Milk of Anatolian Buffaloes. Pakistan Journal of Zoology, 48(2):325-332.

- Şahin, A., Yildirim, A., Ulutaş, Z., 2017. The effects of stage of lactation, parity and calving season on somatic cell counts in Anatolian Water Buffaloes. Indian Journal of Animal Science, 51(1):35-39.

- Şahin, A., Yıldırım, A., Ulutaş, Z., 2019. The Effects of storage temperature and storage time on the somatic cell count of Anatolian Buffaloes. Buffalo Bulletin. 38(2):299-309.

Response:

We thank the editor for attracting our attention to these new publications; we added them to the manuscript. We also enhanced the introduction and the discussion sections to address in more details issues related to the origin and differences of the buffalo species, as well as the adaptability to changes in feed, environment as well as resistance to the disease. 

We reviewed the language throughout the paper and a native speaker helped us improve the English style. 

Response to Reviewers' Comments

Comments to the Author

Reviewer #1: 

1- English style should be revised in the whole manuscript.

Response:

Took into consideration and done.

2- The Ethical Committee information is lost and it should appear in the Material and Methods section.

Response:

Done in the manuscript under materials and methods like suggested. We specifically wrote the following:

“The experiments in this study were performed in compliance with the official decree of the Ministry of Agriculture in Egypt relevant to animal welfare No. 27/1967 regarding the humane treatment of animals (20). Ethics approval was obtained from the Ethics Committees of The Agricultural Genetic Engineering Research Institute and Animal Production Research Institute.”

3- The authors indicate that the analysis performed in this study could provide the identification of a large list of genes related to milk virgin, milk pregnancy, milk lactation, milk involution and milk mastitis, albeit only information about milk lactation related genes is provided in Support Table 3. Please, provide more information about the rest of lists of genes.

Response:

We reviewed support Table (Now it is Support Table 4). The Excel file is composed of multi-sheets (in multi tabs) and a separate sheet for each group of genes is already provided. In addition, we ran pathway and gene ontology analysis, as required in the next comment.

4- Additional graphs about the particular function of the identified genes in each list could help for a better understanding of the genes related to each category. I encourage the authors to make groups of genes that could be involved in different molecular and cellular functions inside each general list using, for example, cheese plots or cheese graphics. 

Response:

We performed Gene Ontology and Pathway analyses using PANTHER pathway kit (Mi, et al., 2019). Each stage of the gene-sets listed in Support Table 4 were classified according to their possible involvement in molecular functions, biological process, cellular component, and different pathways. We also added an extra file (S5 Table.). We also updated the manuscript text and added a new paragraph at the end of the “Results and Discussion” section.

5- The Results and Discussion section needs to be more elaborated and include more references to support the findings.

Response:

Reviewer’s comments to our manuscript were very fruitful. As a result, we added the requested modifications such as more explanation texts, tables, and references. Briefly, the sub-section for comparing our assembly to other available assemblies was enhanced. A new sub-section for pathway and gene ontology was added. Extra sub-sections including conclusions, future works and insights was added to the manuscript.

Reviewer #2: 

What is the application aspect of the study?

Response:

The main outcome of this manuscript is obtaining a high-quality draft of the Egyptian Buffalo Genome. This is considered as a foundation from which further research of the genetic variations in different local breeds of Egyptian Buffalo can be conducted. The information related to this breed, especially the gene lists, along with other buffalo assemblies will enable further research in breeding and milk production, and help in making informed decisions about the directions of their breeding programs.

In the manuscript, we added the following sentences in the conclusion “The Egyptian water buffalo genome offers a unique insight for understanding the variations in sequence between buffalo around the world and cattle milk genes. This variation could help in the future discovery of genes behind complex dairy and meat production traits and to understand the species-specificity of milk composition. Moreover, the availability of various buffalo assemblies with gene annotation will offer a good opportunity for understanding the disease resistance/susceptibility in buffaloes.”

How It is applicable to daily life of farmers?

Response:

Another goal is that this draft assembly will enable large scale variant detection and association studies over Egyptian breeds. We added the following lines in the conclusion section.

“This work will eventually help in identifying and validating specific genes related to the quantitative and qualitative aspects of milk production. Such tools will enable the farmer to predict the characteristics of the desired animal and be able to invest in appropriate animals with higher milk or meat production without wasting effort, time and money.” 

Which sector it covers the most?

Response:

The current manuscript is mainly covering the establishment of Egyptian Buffalo draft genome and therefore is focused on the genetic information, comparative genomics and genome structure. 

Does this study addressed scientists only?

Response:

This study provides a platform for researchers to utilize the genetic information in order to improve the productivity of Egyptian water buffalo breeds.

What is the future of your research?

Response:

Work is currently ongoing to study genetic variations between major Egyptian Buffalo breeds. This will improve our understanding about the genetic makeup of these breeds and facilitate the use of such information in breeding programs. Another direction is to study the correlations between specific markers and qualitative and quantitative traits with the aim of generating a panel of genetic markers that could be used to predict the commercial prospect of each individual animal. Regarding technical aspects, we plan to use long-read sequencing technology in a follow-up project to improve our assembly and produce a chromosome level and well-phased version of it. 

We added following lines in the conclusion “We expect that the next logical step in buffalo genome research is to run large scale genome variation projects on a large number of animals and to conduct association studies to predict genes and variations contributing to the unique characteristics of the buffalo breeds. This step will make use of the available well-phased reference assemblies to efficiently compute these variations.”

Reviewer #3: 

1-My first question when checking the submission of the sequence data is that it was deposited at ncbi the 27-Sep-2013. Since then several water buffalo genomes have being published, the authors mention one (CASPUR-assembly) and use that information to make comparisons. If accepted for publication this work should include the other water buffalo genomes, eg https://doi.org/10.1002/ece3.4965

Response:

We thank the reviewer for this important comment. Yes, many genomes have either been published or even older versions have been updated during the last year. Therefore, we enhanced the section for comparing different breeds. We included all the assemblies deposited in NCBI, including the Bangladeshi buffalo. The list of added assemblies includes also recent assemblies based on long read sequencing for the Mediterranean and African buffalos (please refer to table 3). We also updated the text of the introduction to reflect the availability of more genomes which appeared in the time we were preparing our manuscript. 

I guess that the authors used the cattle as reference genome for the assembly because there was no other water buffalo draft available at the time when this study was initiated? Can the authors explain why they used the cattle and not another water buffalo genome? Will the assembly be different? Discuss the differences.

Response:

We used the cattle genome at that time because it was the only available option. It was assembled at the chromosome level, while the available buffalo assembly was just on the scaffold level with low N50. We were motivated to produce an assembly with better N50. Also the high similarity of both genomes was striking where most of our reads mapped well to the cattle genome (We added this info in Supplementary File 2). This encouraged us to follow that path. In other words, the use of the buffalo draft assembly available at that time would have affected the quality (more gaps and shorted scaffolds) and would lead to lower N50 compared to what we obtained. Our choice turned out to be the best, when high synteny between the cattle and buffalo was later confirmed in the recent publication that updated the Buffalo assembly using long-read sequencing (Low et al 2019); this paper appeared after we finished our manuscript.

In the manuscript, we added sentences in the methodology and results section (page 12) showing our motivation to use the cattle as reference at that time.

We wrote in Page 12: “The cattle genome (Bos_taurus_UMD_3.1.1) was selected as the reference genome. At the chromosomal level, the reads of the Egyptian buffalo covered all chromosomes with an average of 98.1%. Also, the assembled contigs from the mapped reads covered 97.5% of the cattle genome (Supplementary file S2) these primary results justified the use of the cattle genome as a reference for the assembly.”

I recommend to rewrite several parts of the manuscript to make clear what kind of buffalo species and subspecies the authors are referring to in the text (latin names). As example not all the african buffalo belongs to the same sub-species.

doi: 10.1371/journal.pone.0056235

Response: 

Taken into consideration throughout the paper. We also added a paragraph to the Introduction Section (Page 3 of the manuscript) to shed more light on buffalo taxonomy. 

If published I would like to have a table with the different buffalo species mentioned in the article informing about the number of chromosomes that each has. 

Response:

Taken into consideration and a paragraph was added in the Introduction on pages 3 (lines 60-65) of the manuscript. The sub-section comparing different available assemblies was also enhanced with extra meta-data and information. 

Would be very informative to have synteny figures between the Egyptian water buffalo and cattle, African buffalo, other water buffalos.

Response:

The section comparing our assembly to other ones has been re-written to include all other deposited assemblies in the NCBI for Bubalus Bubalis and water buffalo. We also conducted large scale comparison to compute synteny and made a synteny figure showing the synteny of each assembly compared to the reference buffalo and cattle genomes. In this comparison, we used all available assemblies in NCBI for the Bubalus Bubalis and African Buffalo. We made sure that we use the latest version of any updated assembly. We found that the Buffalo UMD genome was updated after the use of long-read sequencing technology and became UOA assembly. We also found that the African genome was also updated with results from long-read sequencing technology. We included all these new updates in the manuscript. 

Of interest for the reader it would be good know what are the differences between the Egyptian, the "Italian" , the chinese and other published water buffalo genomes, a comparison at the genome level is desirable. In all, why the Egyptian water buffalo is worth its own genome publication.

Response:

The Egyptian buffaloes are dual purpose animals used for (milk and meat). Local Egyptian breeds were shown to be more efficient than when crossed with either Italian or Pakistani buffaloes (Fooda et, al., 2011 1&2) under our local conditions, proving that Egyptian breeds are more adaptive to the managerial practices, climatic changes and feed. Also, they more resistance to diseases. We updated the text in the introduction and discussion section with this info.

Most publishers ask for accession numbers for the sequence data produced, I would like to have the workflow used in this work deposited somewhere so it can be studied by the reviewers e.g https://fair-workflows.github.io/project.html. There is a link in the article mentioning that the authors used a workflow using Tavaxy, but I could not find any instructions at the Tavaxy site. https://www.tavaxy.org/ I also visited the site and many links (e.g. https://egyptianbuffalo.org/data) gave --- Not Found; The requested URL was not found on this server.; Apache/2.4.29 (Ubuntu) Server at egyptianbuffalo.org Port 443

Response:

We reviewed the web-site (https://egyptianbuffalo.org/) and provided links to the data (runs and assembly) in the NCBI repository and added also links to the dedicated version of Tavaxy for the assembly of the buffalo genome (https://buffalo.tavaxy.org/). 

We provided information about the workflow implementation in Supplementary File 1. The workflow is now available in the form of a Docker container and it is deposited at Docker Hub. We also updated the web-site and provided a link to the implementation (https://buffalo.tavaxy.org) in Tavaxy and the web-based demo version, where the user can run the assembly workflow for some test data 

(http://demo.buffalo.tavaxy.org/tavaxy/webui/login.html). 

The authors would like to thank the reviewers for their valuable comments which significantly enriched the manuscript.

---

## [Decision Letter · Decision Letter 1]

21 Jul 2020

Sequencing and Assembly of the Egyptian Buffalo Genome

PONE-D-20-01913R1

Dear Dr. El Khishin,

We’re pleased to inform you that your manuscript has been judged scientifically suitable for publication and will be formally accepted for publication once it meets all outstanding technical requirements.

Kind regards,

Arda Yildirim, Ph.D.

Academic Editor

PLOS ONE

Additional Editor Comments (optional):

Thank you for your hard work!

Reviewers' comments:

Reviewer's Responses to Questions

**Comments to the Author**

1. If the authors have adequately addressed your comments raised in a previous round of review and you feel that this manuscript is now acceptable for publication, you may indicate that here to bypass the “Comments to the Author” section, enter your conflict of interest statement in the “Confidential to Editor” section, and submit your "Accept" recommendation.

Reviewer #1: All comments have been addressed

Reviewer #3: All comments have been addressed

2. Is the manuscript technically sound, and do the data support the conclusions?

Reviewer #1: Yes

Reviewer #3: Yes

3. Has the statistical analysis been performed appropriately and rigorously? 

Reviewer #1: Yes

Reviewer #3: I Don't Know

4. Have the authors made all data underlying the findings in their manuscript fully available?

Reviewer #1: Yes

Reviewer #3: Yes

5. Is the manuscript presented in an intelligible fashion and written in standard English?

Reviewer #1: Yes

Reviewer #3: Yes

6. Review Comments to the Author

Reviewer #1: The authors have addressed all the suggested comments. The manuscript has been improved and the findings obtained can be very valuable to better identify quantitative and qualitative traits of milk production in Egyptian buffalo.

Reviewer #3: The authors have done a substantial work in meeting the review comments to the first version of the submitted manuscript.

The authors made the suggested additions in the text and provided additional figures. The authors also made the data and workflow accessible to the public.

7. PLOS authors have the option to publish the peer review history of their article (what does this mean?). If published, this will include your full peer review and any attached files.

Reviewer #1: **Yes: **Ana Cristina Calvo

Reviewer #3: **Yes: **Erik Bongcam-Rudloff

---

## [Editor Report · Acceptance letter]

7 Aug 2020

PONE-D-20-01913R1 

Sequencing and Assembly of the Egyptian Buffalo Genome 

Dear Dr. El-Khishin:

I'm pleased to inform you that your manuscript has been deemed suitable for publication in PLOS ONE. Congratulations! Your manuscript is now with our production department. 

Kind regards, 

on behalf of

Dr. Arda Yildirim 

Academic Editor

PLOS ONE